# An Evaluation of the Nursing Practice Environments in Portuguese Prisons

**DOI:** 10.3390/healthcare13040403

**Published:** 2025-02-13

**Authors:** Vítor da Silva Valente, Tânia Maria Silva Azevedo, Marlene Patrícia Ribeiro, Soraia Cristina de Abreu Pereira, Sandra Rita Pereira Fernandes, António Carlos Lopes Vilela, Letícia de Lima Trindade, Olga Maria Pimenta Lopes Ribeiro

**Affiliations:** 1Directorate-General for Reintegration and Prison Services, 1150-122 Porto, Portugal; vitor.s.valente@dgrsp.mj.pt (V.d.S.V.); tania.m.azevedo@dgrsp.mj.pt (T.M.S.A.); 2Abel Salazar Institute of Biomedical Sciences (ICBAS), University of Porto, 4050-313 Porto, Portugal; ribeiro.marlene.27@gmail.com; 3RISE-Health, 4200-319 Porto, Portugal; ritafernandes@esenf.pt (S.R.P.F.); carlosvilela@esenf.pt (A.C.L.V.); olgaribeiro@esenf.pt (O.M.P.L.R.); 4Tâmega and Sousa Local Health Unit, 4560-136 Penafiel, Portugal; 5Portuguese Red Cross Northern School of Health, 3720-126 Oliveira de Azeméis, Portugal; 6Nursing School of Porto, 4200-072 Porto, Portugal; 7Nursing Department, Santa Catarina State University, Chapecó 88035-901, SC, Brazil; leticia.trindade@udesc.br

**Keywords:** nursing, prisons, working conditions, work environment

## Abstract

**Background/Objectives**: The prison environment is a unique context of professional practice, characterized by specific challenges requiring conditions that ensure both the delivery of tailored care to inmates and the well-being of healthcare professionals. This study analyzes the characteristics of nursing practice environments in Portuguese prisons. **Methods**: A descriptive, exploratory, mixed-methods design, combining quantitative and qualitative approaches was conducted in 30 Portuguese prisons. A non-probabilistic convenience sampling method was used to select participants. Data were collected between June and July 2022 using an online self-administered questionnaire. The Statistical Package for the Social Sciences (SPSS), version 28.0, was used to analyze the quantitative data, while Bardin’s Thematic Analysis was used to examine the qualitative data. The Ethics Committee granted ethical approval for the study, and the Directorate-General for Reintegration and Prison Services provided authorization. **Results:** A total of 77 nurses participated in the study, representing 39.4% of the target population. The dimensions with the lowest mean scores included “Nurses’ participation and involvement in institutional policies, strategies, and management”, “Institutional policy for professional qualification”, “Interdependent practices in professional activities”, and “Systematic assessment of nurses’ performance and supervision”. Participants highlighted the need for improved strategies targeting structural, procedural, and outcome-oriented components. **Conclusions**: Finding fields of weakness can greatly enhance the setting for nursing practices. Priority interventions in Portuguese prisons include the following: enhancing human resources; providing adequate infrastructure and equipment; implementing policies to involve, qualify, and assess nurses’ performance; and defining indicators centered on the safety and quality of care as well as the welfare of professionals.

## 1. Introduction

The global prison population, estimated at approximately 11 million, is predominantly concentrated in countries such as the United States, China, Russia, and Brazil [1]. Overcrowding often causes inhumane and vulnerable conditions, adversely affecting both inmates and the professionals who work in these contexts [1].

The prison environment presents multifaceted challenges, including a high prevalence of infectious diseases, mental health disorders, chronic conditions, and substance abuse among inmates. These issues are often exacerbated by overcrowding and security concerns, which can hinder the provision of high-quality healthcare [2,3].

In Portugal, the Directorate-General for Reintegration and Prison Services (DGRSP) is responsible for developing policies related to crime prevention, enforcement, and social reintegration. Its mission includes ensuring conditions compatible with human dignity and contributing to the defense of social order and peace [4]. The Directorate-General for Reintegration and Prison Services implements this mission through a structured system that includes prisons, regional probation offices, electronic surveillance teams, and educational centers [4].

Order No. 175/2020, published on 24 July [5], classifies prisons based on security level and administrative complexity. It categorizes prisons as special, high, or medium security based on the required security level. It also classifies administrative complexity as high or medium based on factors such as security level, capacity, prison population, regimes, programs, and resources.

High levels of administrative complexity apply to prisons with specialized security units, hospital facilities, or those designated to provide specialized healthcare. This category also includes high-security prisons with capacities exceeding 500 inmates or prisons with more than 200 inmates featuring multiple sentencing regimes, financial centers, or economic activities. The high level of complexity also applies to prisons with a capacity exceeding 200 inmates that implement, during each calendar year, at least one of the following programs: rehabilitation programs targeting specific criminal issues or inmate groups, education and vocational training programs, or health promotion and disease prevention initiatives [5]. High-security prisons do not meet these criteria, and medium-security prisons, often housing up to 200 inmates, fall under the medium management complexity category [5].

The Directorate-General for Reintegration and Prison Services employs a mixed internal organizational model, which includes a matrix structure of multidisciplinary teams. Within this framework, the Competence Center for healthcare Management oversees the coordination and delivery of healthcare for the prison population and young individuals in educational centers [6].

Until 2019, the healthcare model in Portuguese prisons relied on private service providers. However, inefficiencies in this approach led to a shift toward strengthening internal healthcare capacities [4]. Consequently, healthcare began to be provided by health professionals, including doctors, nurses, diagnostic and therapeutic technicians, and operational/medical assistants, under a mixed contracting model: professionals integrated into the staff of the Directorate-General for Reintegration and Prison Services and professionals contracted under a service provision agreement [4].

Regardless of the contractual arrangement, nursing practice in prison settings, as in other contexts, must align with the social mission of the profession. This involves ensuring the quality and safety of the care provided while promoting the well-being of the professionals. Achieving this goal requires strategies to foster a positive professional environment for all stakeholders [1,7,8].

Although prison services exhibit distinctive characteristics compared to other professional settings, the care provided to inmates must respect their individuality. This includes ensuring proper nutrition, promoting physical activity, and maintaining family ties and social networks. The overarching goal of these interventions is to support the social reintegration of inmates upon the completion of their sentences [9].

Nurses’ professional practice in prison settings requires clinical proficiency and the ability to balance ethical and security concerns in often hostile and challenging environments [10]. A substantial body of literature highlights numerous challenges faced by nurses working in prison services, including extended working hours, insufficient organizational support, and ethical dilemmas. These factors contribute to burnout and job dissatisfaction [11]. On the other hand, elements such as a sense of purpose and peer support have been identified as positive influences, provided that adequate working conditions are in place [12,13]. Staff shortages and intimidation—whether from colleagues or inmates—negatively impact the health and well-being of nurses in prison environments [10]. These effects can lead to heightened risks of cardiovascular disease, anxiety, depression, and burnout [14,15].

Alongside the demands faced by nurses on the teams, nurse managers face significant challenges, such as aligning nursing interventions with the strategic goals of prison institutions, ensuring quality and safe care for prisoners, and promoting the well-being of healthcare professionals.

Despite growing interest in nursing practice environments over the past decade—particularly their influence on client outcomes, professional well-being, and institutional performance [7,8]—research on this topic remains particularly limited in the prison context.

In this context, evaluating the nursing practice environments in prisons is essential for identifying areas for improvement and fostering better outcomes in prison healthcare systems.

This study focuses on the characteristics of practice environments aligned with the structure, process, and outcome components of Donabedian’s framework [16], which is essential for improving healthcare quality. The structure component includes organizational factors that support nursing professionals in performing their work and the conditions under which care is provided. The process component relates to executing activities inherent to the design and provision of nursing care, guided by established standards. Finally, the outcome component includes both desirable and undesirable changes in care, impacting patients and nursing professionals alike [17].

This study’s central research question is: “How are nursing practice environments in Portuguese prisons characterized?”. The primary objective is to analyze the characteristics of these environments within the context of the Portuguese prisons.

## 2. Materials and Methods

### 2.1. Design

This study employed a descriptive, exploratory, quantitative, and qualitative approach and was examined 30 Portuguese prisons.

### 2.2. Participants

The target population included all nurses and specialist nurses employed by the Directorate-General for Reintegration and Prison Services, totaling 196 at the time of the study. Participants were selected using a non-probabilistic convenience sampling method. Nurses working under health service provision contracts and those on leave or vacation during the data collection period were excluded. The exclusion criterion related to health service provision contracts was due to their limited monthly shifts in prisons, which made it difficult to assess certain dimensions of the nursing practice environments.

### 2.3. Instruments

Data were collected using a self-administered questionnaire, completed online through Office Forms, and divided into three sections. This questionnaire was completed online using Office Forms and comprised three sections. The first section included close-ended questions designed to capture the participants’ socio-demographic, academic, and professional profiles. The second section incorporated the Scale for Evaluating the Environment of Professional Nursing Practice, a tool specifically developed and validated for the Portuguese population [17] and has previously been used in a study carried out in prison settings. Lastly, the third section featured an open-ended question: “What changes would make your professional environment more conducive to high-quality care and the well-being of nurses?”.

The Scale for the Environment’s Evaluation of Professional Nursing Practice (SEE-Nursing Practice) comprises 93 items distributed across three subscales: the Structure Subscale, the Process Subscale, and the Outcome Subscale. Each item is rated on a five-point Likert scale, where 1 corresponds to “never”, 2 to “rarely”, 3 to “sometimes”, 4 to “often”, and 5 to “always” [17].

The SEE-Nursing Practice Structure subscale includes 43 items divided into six dimensions. Dimension 1—Human Resource Management and Service Leadership comprises 12 items; Dimension 2—Physical Environment and Conditions for Service Delivery, consists of 13 items; Dimension 3—Nurses’ Participation and Involvement in Policies, Strategies, and Institutional Management, includes 8 items; Dimension 4—Institutional Policy for Professional Qualification, contains 3 items; Dimension 5—Organization and Management of Nursing Practice, includes 4 items and Dimension 6—Quality and Safety of Nursing Care, comprises 3 items.

The SEE-Nursing Practice Process comprises 37 items distributed across six dimensions. Dimension 1—Collaboration and Teamwork, includes 9 items; Dimension 2—Strategies for Ensuring Quality in Professional Practice, consists of 7 items; Dimension 3—Autonomous Practices in Professional Practice, also contains 7 items; Dimension 4—Care Planning, Evaluation, and Continuity, includes 6 items; Dimension 5—Theoretical and Legal Support of Professional Practice, is composed of 4 items and Dimension 6—Interdependent Practices in Professional Practice, consists of 4 items.

The SEE-Nursing Practice Outcome includes 13 items divided into two dimensions. Dimension 1—the Systematic Assessment of Nursing Care and Indicators, contains 7 items, while Dimension 2—the Systematic Assessment of Nurses’ Performance and Supervision, contains 6 items.

### 2.4. Data Collection

As described earlier, data collection relied on a self-administered online questionnaire. Following an agreement made during a prior meeting with the nursing director of the DGRSP, a link to the questionnaire was provided and distributed by nurse managers or nurses in coordinating roles. The link reached the institutional email addresses of all nurses in their respective services who were part of the DGRSP nursing staff. Data collection took place between June and July 2022. Completing the questionnaire took approximately 20 to 25 min.

### 2.5. Data Analysis

Quantitative data were analyzed using The Statistical Package for the Social Sciences (SPSS, version 28.0). Descriptive statistics, including means and standard deviations (±), were used for quantitative variables, while absolute and relative frequencies were calculated for categorical variables.

The analysis of the quantitative results related to the nursing practice environment followed the guidelines set by SEE-Nursing Practice [17].

The analysis applied the following criteria to the average scores of the components or dimensions: scores below 1.75 indicated a practice environment that was not favorable to the quality of care and nurses’ well-being; scores between 1.75 and 2.75 indicated a moderately favorable environment; scores between 2.75 and 3.75 indicated a favorable environment; and scores above 3.75 indicated a very favorable environment for both the quality of care and nurses’ well-being [17]. The analysis of the qualitative data from Section 3 of the instrument used Bardin’s Thematic Analysis [18] and relied on Donabedian’s theoretical framework [16]. The analysis followed these steps: (1) collecting participants’ responses to the open-ended question; (2) pre-analysis: reading the responses to organize initial ideas; (3) categorizing the data: reviewing the material to create categories and/or subcategories; and (4) interpretive analysis: examining the categories and subcategories using the literature review [18].

It is important to note that in phase 3—categorizing the data—based on Donabedian’s framework, the most significant findings were initially organized into three categories: strategies to improve the structural component of the nursing practice environment; strategies to improve the process component of the nursing practice environment; and strategies to improve the outcome component of the nursing practice environment. Subsequently, the findings within each category were organized according to the different types of strategies mentioned by the participants, which correspond to the subcategories.

### 2.6. Ethical Approval and Informed Consent

The study received ethical approval from the Ethics Committee of the School of Nursing of Porto on 20 April 2022 (Process 323-2022), and authorization from the Directorate General of Reintegration and Prison Services on 12 May 2022 (Official Letter 101/CCCRE). Informed consent was obtained from all participants, and their anonymity was guaranteed.

## 3. Results

### 3.1. Characterization of the Participants

A total of 77 nurses participated in the study, representing 39.38% of nurses working for the Directorate General of Reintegration and Prison Services. Table 1 summarizes the participants’ sociodemographic and professional characteristics.

### 3.2. Nursing Practice Environments in Prison Services

The SEE-Nursing Practice instrument revealed distinct scores across the three subscales. The Outcome subscale recorded the lowest mean score (2.86 ± 1.32), followed by the Structure subscale (2.91 ± 0.81). In contrast, the Process subscale achieved the highest mean score (3.55 ± 0.66).

Within the structure subscale, the dimension “Nurses’ participation and involvement in policies, strategies, and institutional management” received the lowest mean score (2.41 ± 0.98). Conversely, “People management and service leadership” yielded the highest score (3.44 ± 1.14).

In the process subscale, the dimension “Interdependent practices in professional practice” showed the lowest mean score (2.50 ± 0.69), while the dimension “Autonomous practices in professional practice” achieved the highest mean score (3.75 ± 0.66).

Regarding the outcome subscale, the dimension “Systematic assessment of nurses’ performance and supervision” registered the lowest mean score (2.55 ± 1.39).

Table 2 presents the mean values for all components and dimensions of the SEE-Nursing Practice.

An in-depth analysis of the results highlighted the items with the highest and lowest scores. Table 3 summarizes the ten items with the lowest average scores, which predominantly correspond to the structure component.

In contrast, the items with the highest average scores, detailed in Table 4, are exclusively associated with the process component.

### 3.3. Strategies for Enhancing Nursing Practice Environments Within Prison Services

The content analysis of responses to the question, “What changes to your professional environment could enhance the quality of care and promote nurses’ well-being?” revealed three categories. These categories were based on the structure, process, and outcome components of Donabedian’s framework [16]: strategies to improve the structure component; strategies to improve the process component; and strategies to improve the outcome component.

#### 3.3.1. Strategies for Improving the Structure Component

Participants’ responses regarding structural improvements emphasized aspects of human resources, material resources, and physical conditions, including:

“*Providing quality nursing care with sufficient human resources*”(P66)

“*Improving the nurse/inmate ratio*”(P1)

“*Improving the quantity and quality of computer equipment*”(P8)

“*Providing more and better materials to ensure quality care*”(P30)

“*Improving working conditions in terms of infrastructure*”(P5)

From the participants’ perspective, enhancing working conditions should be a priority for management. They also highlighted the need to acknowledge nurses’ pivotal roles, competencies, and postgraduate training:

“*The significance of nurses in the prison environment*”(P17)

“*The acknowledgment and appreciation of professional competencies*”(P56)

“*The acknowledgment by the institution of nurses’ postgraduate training*”(P14)

Regarding professional careers, participants underscored the importance of establishing a clear framework for nursing roles:

“*In addition to cultivating the distinct expertise of nursing within prison services, establishing a specialized nursing career aligned with the broader National Health System is paramount*”(P56)

Participants also noted issues with professional relationships, particularly precarious contracts, emphasizing the need for stability and fair compensation:

“*It is crucial to have colleagues performing their duties under public employment contracts rather than service contracts*”(P7)

“*Professional ties need to be more effective*”(P12)

“*I have been in my current position for 21 years, yet my salary remains at the initial pay scale. Compensation must be equitable*”(P17)

Additionally, participants stressed the importance of nurse engagement in decision-making processes related to prison policies and fostering closer relationships with management. They also called for quality and safety systems, highlighting the need for quality indicators:

“*Safety control systems and audits*”(P26)

“*Promote professional satisfaction and retention by investing in a safety plan and defining specific satisfaction and quality indicators*”(P36)

The need for a computerized information system was a recurring theme:

“*Standardizing nursing records in prison services by adopting computerized records is urgent*”(P30)

Participants also emphasized the necessity of standardizing practices through protocols and procedures, as well as tailored in-service training:

“*Define action protocols according to evidence*”(P64)

“*An annual training plan aligned with the objectives and needs of the Health Services*”(P44)

The participants highlighted the importance of nurses’ support services and a tailored prison context evaluation system.

“*Supporting professionals going through personal or professional complex phases*”(P62)

“*Adapting the performance evaluation system to the reality of prison services*”(P11)

#### 3.3.2. Strategies for Improving the Process Component

Regarding process-related strategies, participants emphasized the importance of reflection and collaboration within nursing teams, multidisciplinary teams, and other prison services:

“*Reflecting with the nursing team on the care provided*”(P62)

“*Greater interaction with other nurses leads to improved nursing practice*”(P48)

“*Discuss in a multidisciplinary team*”(P62)

“*Regular nursing team and clinical service meetings promote greater collaboration*”(P20)

“*The exchange of professional experiences between different prisons would be important*”(P26)

Participants also highlighted the importance of fair performance evaluations that identify fields for development:

“*Carrying out performance appraisals, since we have no idea what we can improve*”(P48)

“*Fairer performance evaluations*”(P56)

The participants highlighted transparency and adherence to professional regulations:

“*Transparency, responsibility, and professional practice following regulations, competencies, and quality standards*”(P2)

To enhance communication and continuity of care, participants proposed implementing face-to-face shift handovers and case discussions during transitions:

“*Face-to-face shift handovers should be mandatory*”(P35)

“*Discuss clinical cases during shift handovers*”(P59)

Lastly, participants emphasized the need for professional recognition and appreciation:

“*Valuing and recognizing the importance of nurses in the prison environment*”(P17)

“*Valuing the role of nurses within the multidisciplinary team*”(P59)

“*Recognizing daily work and commitment*”(P38)

#### 3.3.3. Strategies for Improving the Outcome Component

In the field “strategies for improving the outcome component”, participants emphasized the critical role of monitoring and evaluation in driving change. Specifically, they highlighted the importance of assessing the quality of care, the work environment, nurses’ performance, satisfaction, and motivation.

Participants provided the following insights:

“*Better evaluation/monitoring of care provided*”(P21)

“*Ability to extract quality indicators to improve the care provided*”(P56)

“*Continuous monitoring of nurses’ performance was essential*”(P56)

“*Implementing strategies to improve and monitor the working environment*”(P77)

“*Defining outcome indicators (…) that reflect the value of the care provided and promote the satisfaction and retention of health professional talent in the DGRSP*”(P36)

“*Management has focused on improving the effectiveness and efficiency of care and promoting the motivation and mental health of nurses (…) aspects that should be evaluated frequently*”(P38)

Table 5 summarizes the categories and subcategories identified.

## 4. Discussion

This study aimed to identify nurses’ perceptions of the nursing practice environment within the Portuguese Prison Service. Regarding sociodemographic and professional characteristics, the participants, in line with data reported by the Order of Nurses [19], had a mean age of 39.3 years. Most were female, married or in a non-marital partnership, and primarily worked as nurses in highly complex prison services. Among the 34 nurses with a specialization, 70.6% (n = 24) specialized in mental health and psychiatric nursing.

This study provides a comprehensive overview of the nursing practice environment in prisons, highlighting key aspects of the structure, process, and outcome which have also been identified in previous studies [16,17]. The process component, suggesting that nurses in the prison context viewed their professional performance positively, has the highest mean scores. In contrast, the lowest scores were associated with the structure component, indicating a need for investment in improving conditions for effective professional practice. None of the items had a mean <1.75, which indicates that none of them were considered unfavorable to the quality of care and the well-being of nurses.

Despite these generally positive findings, the analysis identified facilitating factors and significant barriers within the nursing practice environment. These factors influence the quality of care provided and the well-being of nurses working in the Portuguese Prison Service.

The results across the three subscales showed that structure and process were perceived as favorable to the quality of care and nurses’ well-being, with mean scores of 2.91 ± 0.81 and 3.55 ± 0.66, respectively. In contrast, the outcome component was considered moderately favorable, with a mean score of 2.68 ± 1.32.

Regarding the structural component, although participants rated the dimension “Physical environment and conditions for running services” as favorable to the quality of care, with a mean score of 2.77 ± 0.87, the qualitative findings revealed participants’ concerns about the availability of human and material resources, as well as the adequacy of infrastructure for healthcare provision. The difference between the quantitative and qualitative findings may be due to participants’ emphasis on highlighting the less favorable aspects, as well as the diversity across the different prisons. Key factors contributing to staff burnout include the need for more personnel, adequate materials, and improved facilities. This finding aligns with the results of Keller et al., who also identified high workloads and insufficient resources as factors that can negatively impact nurses’ physical and mental health [14].

Beyond human and material resources, participants underscored the necessity of a computerized information system to streamline documentation processes. In addition to enhancing nurses’ daily workflow and promoting more efficient and safer care [14], according to the qualitative results, participants highlight how these systems can generate indicators to access care needs and improve resource allocation. Shelton et al. also noted that structural challenges, such as paper-based medical records and security issues, could be easily addressed with an electronic record system [20].

Within the structural component, participants also identified the lack of stable employment for all nurses working in prisons and the absence of professional recognition. These issues reflect both the instability of certain employment relationships and dissatisfaction with limited career development opportunities. The dimensions “nurses’ involvement in policies, strategies, and management” and “institutional policies for professional development” received the lowest scores. These findings align with studies conducted in Portugal’s hospital settings [21]. They suggest that, regardless of the professional context, these fields consistently require further investment throughout the country.

The item “The institution presents motivation strategies, as well as reward and incentive to nurses” received the lowest mean score (1.84 ± 1.25). This result is consistent with the qualitative findings, particularly “Strategies to improve the structure component”, where the subcategories “Management’s appreciation of the role of nurses”, “Valuing competencies”, and “Recognition of postgraduate training” were frequently emphasized by participants.

In various professional contexts, experts have identified valuing the role of nurses as a critical strategy for fostering a more positive working environment. This includes emphasizing the active participation of nurses in institutional policies and providing opportunities for professional development. Organizational support, investment in qualifications, and a robust performance evaluation system are essential for success [7,21]. Stephenson and Bell also underscore the importance of organizational support, highlighting their role in improving job satisfaction and retaining nursing professionals [13].

The findings of this study suggest that creating a dedicated nursing career path for prison services could address many of the challenges identified, improving working conditions and professional satisfaction.

In the process component, the dimension “Interdependent practices in professional practice” received the lowest mean score (2.50 ± 0.69), which aligns with the qualitative findings. Participants highlighted the importance of multidisciplinary team reflection, interprofessional collaboration, and sharing experiences with colleagues from other prisons as key strategies for improvement. Previous studies also found that strategies like regular team meetings, reflecting on clinical practice, and sharing experiences across prisons are key to improving work processes [10,11].

Another aspect highlighted by the participants concerns the transmission of information during shift changes. The qualitative findings show that nurses consider the face-to-face sharing of information between shifts and the discussion of clinical cases addressing inmates’ care needs as fundamental factors for improving communication and continuity of care. In this regard, it is also suggested that information transfer during shift changes should be more structured.

Fairness and transparency in performance appraisal processes emerged as critical factors for nurses’ professional and personal development. The qualitative findings emphasized creating a performance appraisal system based on fair criteria to improve performance, increase self-esteem, and encourage professional recognition.

Multicomponent educational interventions utilizing simulated practice have the potential to enhance skill development, improve professional performance, and ultimately improve the quality of care provided in prisons [20].

For the outcome component, nurses emphasized the importance of continuous monitoring of care quality and performance indicators. Such data are essential for identifying shortcomings, adjusting clinical practices, and optimizing available resources. Following the qualitative findings, it became clear that participants believe the definition of indicators adapted to the prison context could provide valuable information for targeted interventions and the formulation of effective public policies.

Shelton et al., while focusing on the American context, cautioned that despite advances in expanding nurses’ roles within the healthcare system, the challenges faced by nursing in the prison environment remain insufficiently addressed [22]. This highlights the need for a comprehensive set of structural, process, and outcome indicators. Continuous monitoring of these indicators would not only improve care quality but also foster greater transparency, accountability, and trust between professionals and various levels of management. When used appropriately, these indicators can drive structural and procedural changes in prison settings, ensuring that care is effective, efficient, and aligned with best practices.

In the outcome component, it is noteworthy that the dimension “Systematic assessment of nurses’ performance and supervision” received the lowest mean score, aligning with qualitative findings. Participants consistently highlighted the need for enhanced performance monitoring and supervision as a key improvement strategy.

To foster a positive nursing practice environment, it is essential to implement mentoring and preceptorship programs that, in addition to promoting the quality of care, ensure greater job satisfaction among nurses [23]. In addition to mentoring and preceptorship programs, researchers have identified interventions, practices, and training programs to improve nursing practice environments [23,24]. Authors categorize interventions aimed at enhancing nursing practice environments into three approaches: accreditation processes, educational strategies, and a participatory approach [24].

Researchers classify interventions designed to meet the criteria set by an accreditation body as accreditation process strategies [24]. Educational strategies focus on enhancing nurses’ skills [24]. Lastly, the participatory approach includes interventions that foster innovation in work settings, improve processes, and promote leadership, teamwork, autonomy, and communication [24].

Interventions using a participatory approach consistently produced positive effects on nurses, clients, and organizations [24]. Since participants in this study stressed the need for more involvement, using interventions with a participatory approach [24] would be effective in the prison setting.

The qualitative findings also highlighted concerns about the mental and emotional health of nurses working in prisons. These professionals face heightened exposure to risks in the prison environment, including physical and psychosocial hazards, which increase their vulnerability to adverse outcomes [14]. Burnout, anxiety, and lack of recognition significantly impact motivation and performance. A Turkish study found that these factors harm work-related quality of life and, in turn, reduce organizational commitment [25].

Keller et al. emphasize that hostile and high-pressure work environments necessitate institutional policies prioritizing the physical and psychological well-being of professionals [3]. Similarly, Karaaslan and Ayfer highlight that it is essential to recognize prisons as important nursing environments and address the needs of nurses within these settings [25]. Based on their study’s findings, they stress the importance of adapting and improving working conditions, offering career advancement opportunities, ensuring a balance between personal and professional life, and placing greater focus on mental health issues affecting nurses [25].

Strategies like psychological support programs, regular supervision, and creating spaces for open dialog within teams are essential for reducing the negative impacts of the prison environment on healthcare professionals. Moreover, implementing mental health programs for nurses—providing seamless access to counseling and emotional support services—has alleviated occupational stress and burnout [3]. Future interventions focused on improving the prison work environment and supporting nurses’ well-being should incorporate strategies to increase job autonomy, reduce stress, ease work demands, and cultivate positive workplace relationships among colleagues [12].

The study’s findings underscore the necessity of a holistic approach to promoting a positive nursing practice environment in prison settings. This includes structural changes and collaborative processes grounded in trust, mutual respect, and effective communication. One must accompany continuous monitoring of care practices with periodic reviews of implemented strategies to ensure their effectiveness and adaptability to dynamic circumstances.

Strengthening institutional policies, valuing healthcare professionals, and creating optimal working conditions for nurses are fundamental steps toward cultivating a positive practice environment. Such efforts are crucial for enhancing the satisfaction and well-being of professionals and improving the quality and safety of healthcare services provided in Portuguese prisons. Developing strong public policies tailored to the unique challenges of the prison context could be key to transforming these environments. Such policies would promote both professional satisfaction and the quality and safety of the care provided.

### Limitations

Despite the relevance of the findings, this study has several limitations. Firstly, the use of convenience sampling for conducting a study aimed at assessing the practice environment in Portuguese prisons. In addition to limiting the generalizability of the results, this can introduce some bias due to the representation of subgroups of participants or their interest and knowledge of the topic, which may motivate participation without representing the studied population. Second, security requirements and limited access to prisons for face-to-face data collection likely influenced the number of nurses who participated by completing the questionnaire online. In addition, as the study was based on self-reports from participants, it is essential to consider the possibility of response bias. Thirdly, it should be considered that nurses on service contracts were excluded, which may have led to the loss of relevant data and, consequently, influenced the results. The diversity of prisons also presents a limitation. Although common guidelines exist, nurses’ working practices and conditions vary widely across prisons due to differences in security levels, infrastructure, and local management. These structural differences may interfere with the identification of factors that, within the practice environments, should be targeted for intervention. In this context, the exclusive participation of nurses in some prisons prevented an assessment of whether perceptions differ between general nurses and specialist nurses regarding practice environments, which constituted another limitation that should be addressed in future studies.

However, the study encourages investment in prison environments to determine whether the weaknesses are common and whether improvement strategies, although adaptable to contexts, can be generalized and replicated. Beyond the national context, conducting studies in prison environments at an international level is essential to understanding the global dynamics that impact healthcare delivery in these settings. The sharing of evidence and best practices among different countries can contribute to the development of more effective and adaptable strategies, fostering improvements in the quality of nursing care in prisons on a broader scale.

## 5. Conclusions

This study assessed the nursing practice environment in Portuguese prisons, revealing that the structure and process components were generally favorable to care quality and nurses’ well-being, while the outcome component was moderately favorable for both aspects.

Although none of the 14 dimensions of the Scale for the Evaluation of the Environment of Professional Nursing Practice were rated unfavorable to the quality of care and nurses’ well-being, some received moderately favorable ratings. These include, within the structure component: “Nurses’ participation and involvement in policies, strategies and running the institution” and “Institutional policy for professional qualification”; within the process component: “Interdependent practices in professional practice”; and within the outcome component: “Systematic assessment of nurses’ performance and supervision”.

The participants emphasized strategies for improving the practice environment, focusing on the structure component. They highlighted the need for increased staffing, better equipment and infrastructure, the implementation of computerized information systems, greater recognition of the nursing role by management, and enhanced professional qualifications. For the process component, they called for urgent measures to improve care transitions and encourage reflection among nurses and the multidisciplinary team on the care provided to inmates. Fair performance evaluation procedures that accurately reflect nurses’ work in prison services showcased their performance. Regarding the outcome component, participants stressed the need to establish systems for monitoring care quality, the practice environment, and professional satisfaction. Other key aspects included ensuring stable, permanent employment relationships and offering fair compensation that aligns with nursing roles and career development.

This evaluation identifies the weakest fields within the nursing practice environment in prisons and highlights key interventions that can significantly enhance the professional practice conditions for nurses in Portuguese prisons.

## Figures and Tables

**Table 1 healthcare-13-00403-t001:** Participants’ Sociodemographic and Professional Characterization.

Sociodemographic and Professional Characteristics	
Gender n (%)	
Female	52 (67.5)
Male	25 (32.5)
Marital Status n (%)	
Married/non-marital partnership	45 (58.4)
Single	26 (33.8)
Divorced	5 (6.5)
Widower	1 (1.3)
Age (years) Mean; Std. Dev.	39.3; ±10.5
Education n (%)	
Bachelor’s degree	62 (80.5)
Master’s degree	15 (19.5)
Professional title n (%)	
Nurse	56 (72.7)
Specialist nurse	21 (27.3)
Nursing specialization n (%)	
Psychiatry and mental health	24 (70.6)
Medical–surgical	3 (8.8)
Community and public health	3 (8.8)
Rehabilitation	2 (5.8)
Medical–surgical in critical care	1 (2.9)
Family Health	1 (2.9)
Prison service typology n (%)	
High grade	42 (54.5)
Medium grade	30 (39.0)
Other	5 (6.5)

Std. Dev.—Standard deviation.

**Table 2 healthcare-13-00403-t002:** Mean Values for the Components and Dimensions of Nursing Practice Environments.

Components and Dimensions of Nursing Practice Environments	Mean	Std. Dev.
Structure Component	2.91	0.81
Dimension 1—People management and service leadership.	3.44	1.14
Dimension 2—Physical environment and conditions for service running.	2.77	0.87
Dimension 3—Nurses’ participation and involvement in policies, strategies and running the institution.	2.41	0.98
Dimension 4—Institutional policy for professional qualification.	2.44	0.96
Dimension 5—Organization and guidance of nursing practice.	3.07	1.05
Dimension 6—Quality and safety of nursing care.	2.92	1.39
Process Component	3.55	0.66
Dimension 1—Collaboration and teamwork.	3.54	0.80
Dimension 2—Strategies for ensuring quality in professional practice.	3.53	1.33
Dimension 3—Autonomous practices in professional practice.	3.75	0.66
Dimension 4—Care planning, evaluation and continuity.	3.36	0.79
Dimension 5—Theoretical and legal support of professional practice.	3.81	0.78
Dimension 6—Interdependent practices in professional practice.	2.50	0.69
Outcome Component	2.68	1.32
Dimension 1—Systematic assessment of nursing care and indicators.	2.78	1.38
Dimension 2—Systematic assessment of nurses’ performance and supervision.	2.55	1.39

Std. Dev.—Standard deviation.

**Table 3 healthcare-13-00403-t003:** SEE-Nursing Practice Items with Lower Mean Scores.

Item	Mean (Std. Dev.)	Component
Nurses’ workload is systematically monitored	2.39 (±1.58)	Outcome
The institution promotes the participation of nurses in commissions/work groups in the context of continuous quality improvement	2.30 (±1.40)	Structure
The institution promotes the participation of nurses in the definition of internal policies	2.29 (±1.29)	Structure
Nurses’ professional satisfaction is systematically monitored	2.29 (±1.52)	Outcome
The practice of nurses is fundamentally centered on the management of signs and symptoms of the disease	2.26 (±1.42)	Process
The institution provides specialized services to nurses who face problematic situations	2.22 (±1.52)	Structure
Information and communication technologies are suited to the needs of the service	2.09 (±1.15)	Structure
The institution has a policy of encouraging innovation and research in nursing	2.09 (±1.47)	Structure
The institutional training policy considers the training needs of nurses	2.05 (±1.08)	Structure
The institution presents motivation strategies, as well as reward and incentive to nurses	1.84 (±1.25)	Structure

Std. Dev.—Standard deviation.

**Table 4 healthcare-13-00403-t004:** SEE-Nursing Practice Items with Higher Mean Scores.

Item	Mean (Std. Dev.)	Component
The information transmitted in the shift promotes continuity of care in subsequent shifts	4.32 (±0.98)	Process
The information transmitted during the shift change is specific to nursing	4.16 (±0.96)	Process
The strategies adopted for the shift change, such as the duration and location, are appropriate to ensure continuity of care	4.09 (±1.17)	Process
Nurses are concerned with valuing autonomous interventions	3.99 (±0.68)	Process
In professional practice, nurses value knowledge in the field of nursing	3.97 (±0.81)	Process
Nurses act in accordance with the regulatory instruments of professional practice	3.95 (±0.87)	Process
In clients with recovery potential, the practice of nurses is centered on the reconstruction of autonomy	3.94 (±0.80)	Process
When delegating tasks to functionally dependent professionals, nurses carry out appropriate supervision	3.92 (±1.90)	Process
The working relationship between doctors and nurses facilitates assistance to clients	3.86 (±1.20)	Process
The nurses’ clinical opinion is considered when planning the discharge of clients	3.84 (±1.45)	Process

Std. Dev.—Standard deviation.

**Table 5 healthcare-13-00403-t005:** Strategies for Improving the Nursing Practice Environment in Prisons.

Component	Strategies for Improving the Nursing Practice Environment
Structure Component	- Availability of human and material resources - Adequate physical conditions - Valuing the role of nurses - Valuing nurses’ skills - Recognition of postgraduate training - Clear definition of the nursing career - Stability of contract - Opportunity to participate in prison policies - Closeness between management and nurses - Existence of a system to guarantee the quality and safety of care - Existence of a computerized information system - Existence of protocols and procedures to standardize practices - Existence of in-service training - Existence of support services for nurses - Existence of a performance evaluation system
Process Component	- Reflection on care - Interprofessional collaboration - Fair and development-promoting performance evaluation - Acting by the regulatory instruments for professional practice - Adequate transmission of information - Need for professional recognition and appreciation
Outcome Component	- Monitoring the care provided - Monitoring the working environment - Monitoring nurses’ performance - Monitoring job satisfaction - Monitoring professional motivation

## Data Availability

The data that support the findings of this study are available from the corresponding author upon reasonable request.

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
