# Peer review of "An Evaluation of the Nursing Practice Environments in Portuguese Prisons"

_healthcare, 2025, doi:10.3390/healthcare13040403_

Round 1

Reviewer 1 Report

Comments and Suggestions for Authors

I read with interests the paper titled "An Evaluation of the Nursing Practice Environments in Portuguese Prisons"

I have some comments that could enhance the readability and presentation of the paper. 

1. I believe the introduction will gain to be rephrased and reorganized. The information is there, but some sentences are not connected with previous ideas. Eg1: Line 62, 63 could be connected with previous idea. Eg2 - the beggining of sentence in line 53, 64, 84 could be enhanced, since the first information is sometimes too long and doesn't add much to the context explained. Proofreading by a native speaker is also expected. 

2. Please further explain participant selection. If a convenience sample was used, then no participant selection was made; rather authors include as much as they can within the period of the study. 

3. A drawback that should be further discussed in that authors did not include "temporary employment contracts". As far of my knowledge, most of the prisons in Portugal used to have contract enterprises for healthcare professionals, meaning that the contracts were temporary, but often the nurses worked for years (decades) in the same institution with full-time "temporary" contracts. Commonly, the nurse team within a prison could be predominantly composed by temporary contracts, meaning that authors could lost a lot of data with that exclusion criteria. - clarify rhat do you mean by "temporary contract". 

4. Acronyms as DGRSP are defined multiple times in the manuscript. 

5. The authors state that the reporting of the study was guided by the Strengthening the Reporting of Observational Studies in Epidemiology (STROBE®) tool. This is, however, the major drawback of the paper, since out of 22 items of STROBE, the paper fails to present multiple items. If the authors really want to follow STROBE, it means that the report (final manuscript), should come as described in the strobe guidelines. 

Comments on the Quality of English Language

The paper could benefit from scientific English editing or proofreading by a native English speaker. 

Author Response

Dear Reviewer,

We would like to thank you for the opportunity to improve our article. The suggestions are very relevant and important to us.

The revised version was prepared and approved by all the authors. Please note that the changes are highlighted in yellow in the main document.

The responses to requests are presented in the table in attached file.

Reviewer 2 Report

Comments and Suggestions for Authors

Dear Authors,

Based on my review of the manuscript, I suggest the following points for the authors to address.

1) While the study employs both quantitative and qualitative approaches, there is insufficient justification for the sampling strategy. The convenience sampling method limits generalizability, especially in a specialized setting like prisons. The authors should elaborate on how the sample reflects the broader nursing population in Portuguese prisons or discuss the implications of this limitation more thoroughly. Strengthening the rationale for the sample size or including additional context on participant diversity could enhance the robustness of the findings. Please, improve this important aspect.

2) I detected many self citations attributed to different co-authors. Please, reduce them.

3) The abstract need to be improved. While an attractive text is required, it lacks of originality and clarity.

4) The statistical methods applied (e.g., descriptive statistics and Bardin’s Thematic Analysis) are briefly described but lack detail. For example, there is minimal explanation about how quantitative thresholds (e.g., score ranges for favorable conditions) were established. Adding a supplementary section on the validation of these thresholds or a deeper explanation of how qualitative themes were derived would improve clarity and reproducibility.

5) The "Outcome Component" scored the lowest among the evaluated areas, yet the discussion does not sufficiently address actionable solutions to improve systematic performance evaluation and supervision. The authors could propose specific interventions, such as performance review frameworks or best practices adopted from other healthcare settings, to address these deficiencies.

6) While the discussion highlights structural and process-related issues, there is a lack of depth in linking these findings to broader implications for prison healthcare systems. For example, how could the adoption of a computerized nursing system align with national healthcare priorities? Expanding on how the results could influence policy changes or resource allocation at the national level would enhance the impact of the study.

7) Don't use the third person in the method section (e.g., the authors used a self-administered questionnaire).

8) Please revise them according to MDPI style. References of the last 5 years are preferred, and "ghost references" should be removed.

9) The manuscript's language and structure suggest the potential use of artificial intelligence (AI) tools, such as for drafting, editing, or summarizing content. I detected redundancy of key phrases, and the text maintains an unusually consistent tone. Additionally, while the manuscript discusses findings and implications, it sometimes lacks specific examples or context-specific nuance that a human author would typically include. If AI tools were used in preparing this manuscript, I recommend that the authors disclose this in line with ethical publishing practices. Could you clarify whether AI assistance was utilized, and if so, provide a brief statement in the manuscript detailing how it was used and the extent of human oversight? Transparency in this regard will enhance the manuscript's credibility and adherence to ethical guidelines.

Author Response

(The authors gave the same response as above.)

Round 2

Reviewer 2 Report

Comments and Suggestions for Authors

Dear Authors, compliments for your work and improvement. The manuscript is much improved, with significant strengths in its focus. However, there are still a few areas requiring attention. Specifically:

1) The mixed-methods design should better integrate quantitative and qualitative results to provide a cohesive narrative. The Discussion section often treats them separately, which might reduce the impact of insights from triangulation.

2) The use of non-probabilistic convenience sampling is a limitation that needs clearer acknowledgment and discussion in the limitations section.

3) Although the SEE-Nursing Practice scale is well described, its validation process for this specific context (prison settings) is not discussed, which may raise concerns about its appropriateness.

4) Bardin's Thematic Analysis is mentioned, but examples or excerpts illustrating how themes were derived from qualitative data are absent. Including these would improve transparency and reader trust.

5) The abstract is well structured but could benefit from including one or two sentences on the implications for practice or policy.

6) The reference formatting is inconsistent (e.g., some DOIs are included, while others are not). Standardize the references to align with the journal's requirements.

7) The discussion should elaborate more on how findings could influence policy changes in prison healthcare environments.

An attention to these issues would improve the scientific soundness of your work. Thank you.

Author Response

Dear Reviewer,

We would like to thank you for the opportunity to improve our article. The suggestions are very relevant and important to us.
The revised version was prepared and approved by all the authors. Please note that the changes are highlighted in yellow in the main document. The responses to requests are presented in the attached document.
